# Field-induced ultrafast modulation of Rashba coupling at room temperature in ferroelectric α-GeTe(111)

Geoffroy Kremer [1,2,3] ✉, Julian Maklar [4], Laurent Nicolaï [5,10], Christopher W. Nicholson [1,4,10], Changming Yue[1], Caio Silva[4], Philipp Werner [1], J. Hugo Dil [6,7], Juraj Krempaský[6], Gunther Springholz[8], Ralph Ernstorfer[4,9], Jan Minár [5] ✉, Laurenz Rettig [4] & Claude Monney [1]

Rashba materials have appeared as an ideal playground for spin-to-charge conversion in prototype spintronics devices. Among them, α-GeTe(111) is a non-centrosymmetric ferroelectric semiconductor for which a strong spin-orbit interaction gives rise to giant Rashba coupling. Its room temperature ferroelectricity was recently demonstrated as a route towards a new type of highly energy-efficient non-volatile memory device based on switchable polarization. Currently based on the application of an electric field, the writing and reading processes could be outperformed by the use of femtosecond light pulses requiring exploration of the possible control of ferroelectricity on this timescale. Here, we probe the room temperature transient dynamics of the electronic band structure of α-GeTe(111) using time and angle-resolved photoemission spectroscopy. Our experiments reveal an ultrafast modulation of the Rashba coupling mediated on the fs timescale by a surface photovoltage, namely an increase corresponding to a 13% enhancement of the lattice distortion. This opens the route for the control of the ferroelectric polarization in α-GeTe(111) and ferroelectric semiconducting materials in quantum heterostructures.

Intense femtosecond (fs) light pulses present a powerful tool to control the physical properties of solids, promoting new emergent phenomena inaccessible in the ground state[1–4], as evidenced in ferroelectric (FE) materials such as perovskites[5]. In the family of ferroelectrics, α-GeTe(111) is a fascinating material because it exhibits a giant FE distortion below $T_C \approx 700K$, leading to spin-polarized bulk and surface-split electronic states with the largest Rashba parameter so far reported[6–8]. Based on the inverse spin Hall and inverse Rashba-Edelstein effects, Rashba systems can be efficiently used for spin-to-charge conversion in spintronics devices[9–12]. α-GeTe(111) has been identified as a promising candidate in that direction but, more interestingly, the manipulation of its FE polarization at room temperature is

[1]Département de Physique and Fribourg Center for Nanomaterials, Université de Fribourg, CH-1700 Fribourg, Switzerland. [2]Université Paris-Saclay, CNRS, Centre de Nanosciences et de Nanotechnologies, 91120 Palaiseau, France. [3] Institut Jean Lamour, UMR 7198, CNRS-Université de Lorraine, Campus ARTEM, 2 allée André Guinier, BP 50840, 54011 Nancy, France. [4]Fritz Haber Institute of the Max Planck Society, Faradayweg 4-6, 14195 Berlin, Germany. [5]New Technologies-Research Center University of West Bohemia, Plzen, Czech Republic. [6]Photon Science Division, Paul Scherrer Institut, CH-5232 Villigen, Switzerland. [7]Institute of physics, Ecole Polytechnique Fédérale de Lausanne, CH-1015 Lausanne, Switzerland. [8]Institut für Halbleiter-und Festkörperphysik, Johannes Kepler Universität, A-4040 Linz, Austria. [9]Institut für Optik und Atomare Physik, Technische Universität Berlin, Straße des 17. Juni 135, 10623 Berlin, Germany. [10]These authors contributed equally: Laurent Nicolaï, Christopher W. Nicholson. ✉e-mail: geoffroy.kremer@univ-lorraine.fr; jminar@ntc.zcu.cz

of great interest for next generation non-volatile memory devices with low power consumption. The control of the FE polarization could be used as a knob for controlling the spin-to-charge current conversion sign[12]. Using fs light pulses to manipulate the FE polarization state in $\alpha$-GeTe(111) thus emerges as an exciting perspective for a drastic increase of the performance of the future generation of spintronics devices. Here, using fs extreme-ultraviolet pulses delivered by a table-top high-harmonic-generation (HHG) source[13], we perform time and angle-resolved photoemission spectroscopy (tr-ARPES) as schematically depicted in Fig. 1a. It is a direct and comprehensive technique to track the momentum and energy resolved dynamical evolution of the electronic band structure which allows us to reveal a room temperature photoinduced enhancement of the Rashba coupling that directly signifies an increase of the ferroelectricity in $\alpha$-GeTe(111) on the sub-picosecond timescale.

## Results

### Static investigation

Figure 1 b presents the band structure of thin films of $\alpha$-GeTe(111) along the $\bar{\Gamma} - \bar{K}$ high-symmetry direction, revealed by static ARPES measurements using only the probe pulses, in agreement with existing literature[6–8,14–16]. The observed band structure is in excellent agreement with calculations using the one-step model of photoemission[17] (Fig. 1c) performed for a Te terminated surface with short bonds between the last two atomic lattice planes of the top Te surface and the next sub-surface Ge plane (right part of Fig. 1a). In the vicinity of the Fermi level

($E_F$), the Rashba-split bulk states $B_1$ and $B_2$ are well reproduced (see also second derivative in Supplementary Fig. 1a). Similarly, two surface-derived states labelled $SS_1$ and $SS_2$ are well-resolved. Our surface is well ordered with Te termination and short surface Te–Ge bonds, as evidenced by the surface state dispersion[18], a crucial point for the rest of the discussion. This particular termination leads to a strong outward FE polarization which is at the origin of the large Rashba splitting of the bulk states.

### Out-of-equilibrium dynamics and theoretical investigation

We next investigate the out-of-equilibrium ultrafast dynamics of $\alpha$-GeTe(111) using a stroboscopic pump-probe approach. To do so, we excite the material with infrared (IR) pulses (1.55 eV pump pulses) and examine the response of the system by probing its electronic band structure with the 21.7 eV probe pulses. Figure 2a, b shows the ARPES maps taken at time delays of −200 and +200 fs, respectively. The most striking observation is the transient population of the Rashba-split surface state $SS_1$ and $SS_2$ above $E_F$, which perfectly follows the transient elevation of the electronic temperature and the related Fermi-Dirac distribution broadening (see Supplementary Fig. 2). This is confirmed by looking at the difference ARPES map in Fig. 2c.

The red color above $E_F$ corresponds to an increase of the photoemission intensity whereas the blue is associated to a decrease, confirming the transient Fermi-Dirac distribution broadening on this

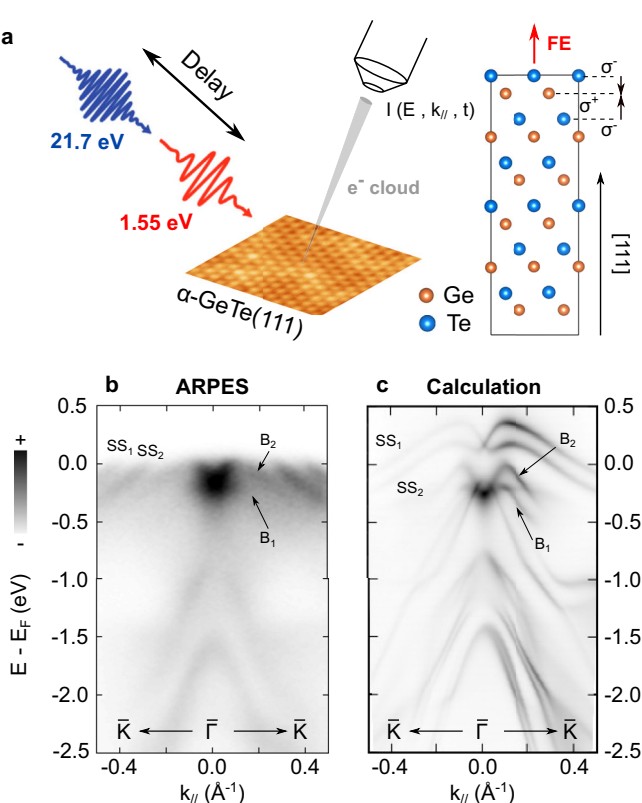

**Fig. 1 | tr-ARPES configuration and static ARPES characterization. a** tr-ARPES experiment using 21.7 eV probe and 1.55 eV pump pulses (left), and surface structure of $\alpha$-GeTe along the [111] crystallographic direction with Te termination and short surface bonds (right). Interatomic dipoles (black arrows) due to negative ($\sigma^-$) and positive ($\sigma^+$) net charges give rise to a net outward FE polarization (red arrow). **b** Static ARPES spectrum along the $\bar{K} - \bar{\Gamma} - \bar{K}$ high-symmetry direction (see second derivative in Supplementary Fig. 1) and **c** corresponding one-step model of photoemission calculations for a Te-terminated surface of $\alpha$-GeTe(111) with short surface bonds.

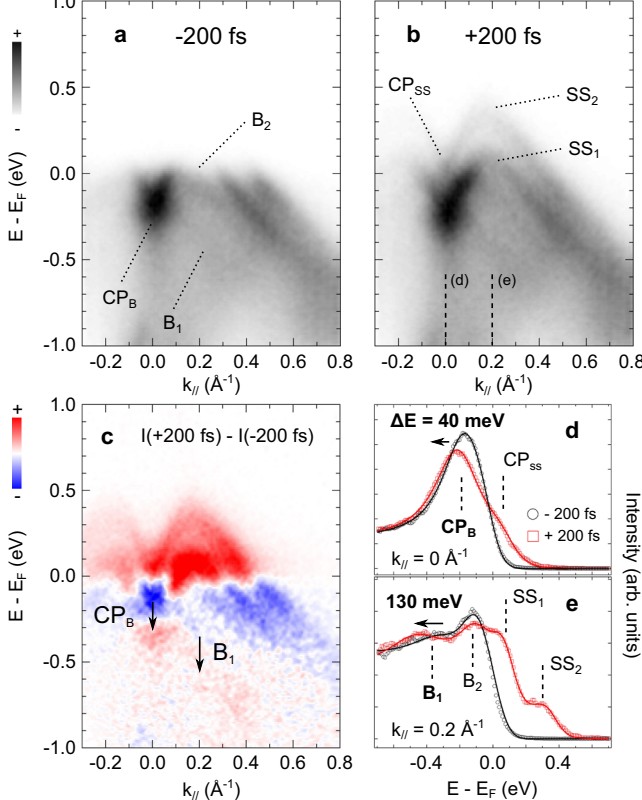

**Fig. 2 | Out-of-equilibrium tr-ARPES measurements. a** tr-ARPES spectra of $\alpha$-GeTe(111) recorded at pump-probe delays of **a** −200 fs and **b** +200 fs. The data have been acquired along the $\bar{K} - \bar{\Gamma} - \bar{K}$ direction with an absorbed pump fluence of 1 mJ/cm². **c** Difference plot between panels **b** and **a**. Red and blue colors correspond to an increase and a depletion of photoemission intensity, respectively. The two arrows highlight the transient shift of the crossing point of the bulk states at normal emission (CP$_B$) and the $B_1$ contribution. This shift is visible as a red/blue contrast on the difference map. Energy distribution curves (EDCs) at −200 fs (black) and +200 fs (red) taken at **d** $k = 0$ Å$^{-1}$ (normal emission) and **e** $k = 0.2$ Å$^{-1}$ (off-normal emission) : see vertical dashed lines in panel **b**. Solid lines correspond to a fit of the raw data.

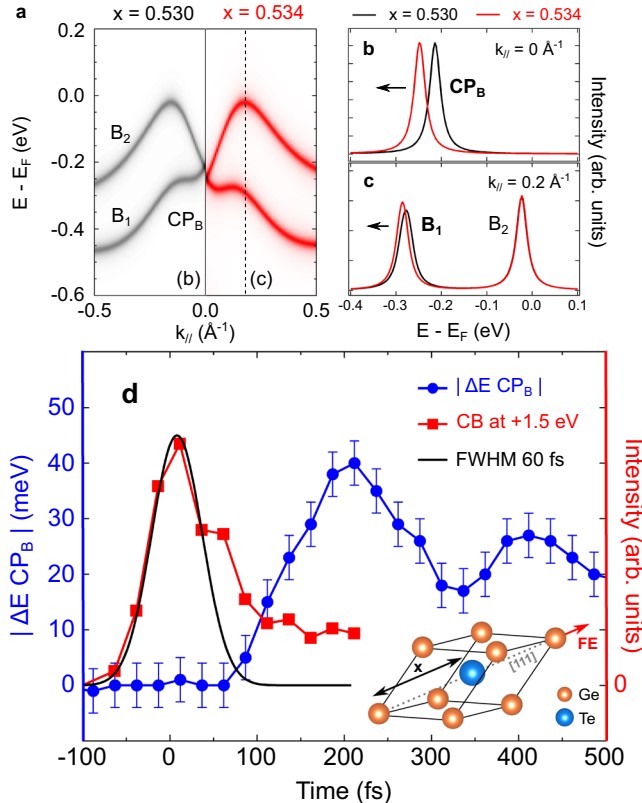

**Fig. 3 | Band structure calculation and temporal evolution of $CP_B$ position.**
**a** Bulk band structure of $\alpha$-GeTe(111) along the $\bar{\Gamma} - \bar{K}$ high-symmetry direction for two different x positions of the Te atom, as defined in the inset of panel **d**. Corresponding EDCs for $x = 0.530$ (black) and $x = 0.534$ (red) taken at **b** $k = 0.0$ Å$^{-1}$ and **c** $k = 0.2$ Å$^{-1}$. **d** Temporal evolution of $CP_B$ extracted from tr-ARPES measurements taken at an absorbed pump fluence of 1.5 mJ/cm$^2$ (blue curve; the error bars have been extracted from the fitting procedure). The red curve corresponds to the transient population of the conduction band and the black curve to the corresponding Gaussian fit.

timescale and the thermal population of the surface state. Close inspection of the difference map in Fig. 2c reveals an additional transient feature. Indeed, a blue/red contrast is also visible in the bulk states region, approximately 400 meV below $E_F$. Here, the red shading (increase) is located at higher binding energy compared to blue ones (decrease), meaning that this intensity change is not related to the above mentioned Fermi edge broadening. Instead, it rather corresponds to a transient shift of the bulk states to high binding energy. We evidence this in Fig. 2d,e with energy distribution curves (EDCs) taken at $k = 0$ Å$^{-1}$ (normal emission) and $k = 0.2$ Å$^{-1}$ (off-normal emission). They show a transient shift of the crossing point of the bulk states ($CP_B$) and $B_1$ contribution after 200 fs, respectively, equal to 40 and 130 meV (the $B_2$ contribution does not shift). Since the Rashba splitting is proportional to the energy difference between the $B_1$ and $B_2$ branches, this is a clear evidence for a transient light-induced increase of the Rashba coupling in $\alpha$-GeTe(111) on the subpicosecond timescale. We have experimentally extracted the transient evolution of the Rashba coupling by estimating the Rashba parameter of the bulk bands at time delays of −200 fs and 200 fs (see Supplementary Fig. 3). The Rashba parameter is given by the relation $\alpha_R = 2E_R/k_0$, where $E_R$ is the energy difference between the $CP_B$ and the top of the $B_2$ branch, and $k_0$ their momentum difference. At time delays of −200 fs and 200 fs, it equals 3.1 eV.Å and 3.8 eV.Å respectively, corresponding to an increase of 18% of the Rashba coupling.

To explain the structural origin of our observation in the electronic structure, Fig. 3a shows band structure calculations around the $\bar{\Gamma}$

point of the BZ for two distinct FE distortions, where we only show the bulk contributions for the sake of clarity. The left panel corresponds to the ground state of the system where the x position of the Te atom is 0.530, expressed as a fraction of the distance between two Ge atoms along the [111] direction (see Fig. 3d inset). Alternatively, the right panel in Fig. 3a shows the case where the FE distorsion has increased to $x = 0.534$ which corresponds to a 13% increase in the FE polarization. As evidenced by Fig. 3b, c, such an increase of ferroelectricity leads to a shift of $CP_B$ at the $\bar{\Gamma}$ point and, at the same time, for higher momenta to bulk band shift $B_1$ to higher binding energy. This is in good qualitative agreement with our experimental observations shown in Fig. 2 and consequently confirms our interpretation since it reproduces well the 40 meV shift of the $CP_B$ we experimentally observe.

As x approaches 0.5, the system tends towards a more centrosymmetric structure (rock salt structure, space group $Fm\bar{3}m$) and consequently to the more paraelectric phase with a less important Rashba coupling (see Supplementary Fig. 4 and Supplementary Fig. 5). Consequently, the ferroelectricity transiently increases, which is highly unusual, as photoexcitation typically leads to a more symmetric phase[19,20].

As a matter of fact, the energy position of $CP_B$ is proportional to the magnitude of the FE distortion in $\alpha$-GeTe(111). To obtain the temporal dynamics of the ferroelectricity induced by photoexcitation, we consequently plot in Fig. 3d the time evolution of the $CP_B$ shift, $\Delta E\ CP_B$, as extracted from our tr-ARPES measurements. This evolution displays two remarkable dynamical features. Firstly, it takes about 80 fs before a $CP_B$ shift occurs. The position of time zero has been obtained from the transient electronic population 1.5 eV above $E_F$ in the conduction band (red curve). It corresponds to the first observable signal due to the optical excitation of electrons from the top of the valence band to the conduction band. In these high energy states, the lifetime of the photoexcited electrons is very short and the corresponding trace almost corresponds to the cross-correlation between the pump and the probe pulses (black gaussian curve). The asymmetric shape of the red curve is associated to a non-negligible lifetime with some exponential decay. Secondly, after 200 fs the $CP_B$ shift reaches its maximum of 40 meV, followed by a coherent oscillation with a period of approximately 200 fs, corresponding to a frequency of 5 THz (see also Supplementary Fig. 6). Thus, the effect of the IR pump pulse is to photoinduce on the sub-ps timescale a coherent modulation of the ferroelectricity in $\alpha$-GeTe(111). Due to the coupling of Bloch states to this FE $A_{1g}$ mode along the [111] direction of the crystal[21], we thereby observe a resulting modulation of the Rashba coupling as probed by our tr-ARPES measurements.

The observations presented above are in notable disagreement with previous literature. Indeed, experimental works using ultrafast electron diffraction[22] and time-resolved X-ray diffraction[23] observed instead that an IR pump pulse produces a successive FE to paraelectric, and a paraelectric to amorphous phase transition in $\alpha$-GeTe. These results were supported by time-dependent DFT calculations suggesting that even in the lowest fluence regime, the effect of an IR pulse on $\alpha$-GeTe is a diminution of the ferroelectricity caused by a reduction of the Ge−Te bonds length[24,25]. Last but not least, we observe an unexpected delay of 80 fs before the coherent oscillation sets in. All these observations are unusual in a standard displacive excitation of a coherent phonon (DECP) picture.

## Discussion
To understand this discrepency, we have to consider that ARPES probes mainly the surface of $\alpha$-GeTe(111), unlike the bulk sensitive previous experimental studies cited above. Figure 4a summarizes the most important steps of the temporal evolution of the ferroelectricity in $\alpha$-GeTe(111) after photoexcitation, which scales with the x position of the Te atom along the $\alpha$-GeTe [111] crystallographic direction (vertical black axis). At the same time the transient shift of the $CP_B$ energy position is

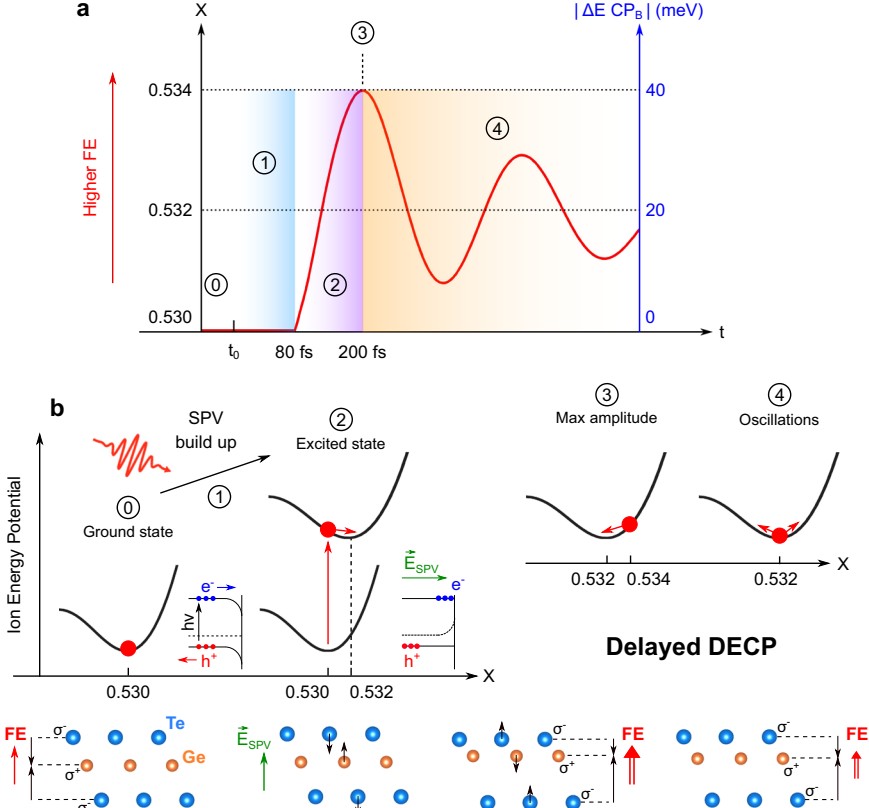

**Fig. 4 | Illustration of the delayed DECP mechanism. a** Schematic evolution of the temporal evolution of the x position of the Te atom in the primitive unit cell of $\alpha$-GeTe(111) and the corresponding variation of the position of $CP_B$ as measured with ARPES. The five steps of the mechanism are numbered from ⓪ to ④. **b** Evolution of the subsurface structure and potential of $\alpha$-GeTe(111) during these five steps. The IR pulse produces a SPV leading to a transient increase of the lattice distorsion and consequently of the ferroelectricity.

displayed on a blue vertical axis, which relates to the experimental blue lineout in Fig. 3d. The five steps of the microscopic mechanism behind the dynamics evolution is detailed in Fig. 4. ⓪ At negative delays, we probe the ground state of the system where the minimum of the ion energy potential corresponds to $x = 0.530$. ① Within the first 80 fs after photoexcitation, a surface photovoltage (SPV) builds up and modifies the electronic energy level diagram of the system. SPV is a well known phenomenon in semiconductors physics[26,27]: it corresponds to a photoinduced spatial redistribution of charge carriers and to a compensation of the pre-existing band bending (BB) in the surface charge region (SCR). Evidence of a positive SPV in our case is given in Supplementary Fig. 7, where we show the pump induced shift to high kinetic energy of the whole ARPES spectrum. As explained in Supplementary Fig. 8, a positive SPV corresponds to a migration of electrons to the surface and holes to the bulk, creating a compensative electric field in the direction out of the sample surface. The SPV build up typically takes a few tens to a few hundreds of fs[26,27]. We performed drift-diffusion calculations on a GeTe surface (see Supplementary Fig. 9 and Supplementary Software 1), demonstrating that the pump-pulses completely suppress the initial BB after a few tens of fs, in excellent agreement with our experimental findings. ② The compensative electric field plays an additionnal and crucial role since it influences the subsurface ferroelectricity. As depicted in this second step of Fig. 4b, the Te atoms are pushed into the bulk and Ge atoms towards the surface, leading to an increase of the FE distortion. Thus, the system is driven into a transient out-of-equilibrium state in which the new ion energy potential minimum is shifted to higher x with respect to the ground state. ③ After 200 fs, the ion displacement reaches the maximum amplitude corresponding to $x = 0.534$, associated to the 40 meV

shift of $CP_B$, as evidenced by our tr-ARPES measurement at +200 fs. ④ Finally, the system oscillates around its new potential minimum localized at an intermediate position of $x = 0.532$ with concomitant periodic oscillation of the ferroelectricity that correlates with the $CP_B$ position and Rashba splitting. As a whole, these five steps explain the experimental behaviours and appear as a delayed DECP mechanism.

In this picture, we understand the light-induced modulation of ferroelectricity in $\alpha$-GeTe(111) as a surface phenomenon. It is compatible with the rapid fluence saturation of the $CP_B$ shift that we observe (see Supplementary Fig. 10), since it is known that the SPV typically saturates in a very low fluence regime[28]. The non-negligible SPV build up time is also consistent with the delay we observe in the $CP_B$ shift. Similar optical mediated manipulation of Rashba coupling has been recently reported in alkali doped 3D topological insulator[29].

## Conclusions and perspectives

To conclude, we have demonstrated ultrafast light-induced modulation of ferroelectricity in the Rashba semiconducting material $\alpha$-GeTe(111) with tr-ARPES measurements. We observed a remarkable delayed enhancement of the Rashba coupling that can be understood by invoking a delayed DECP mechanism originating from the creation of an SPV on the fs timescale inside the sub-surface region. Beyond its implications for the fundamental research adressing the out-of-equilibrium dynamics of semiconducting ferroelectric materials, our study validates the use of fs light pulses for the manipulation of the polarization in $\alpha$-GeTe(111) at room temperature and opens promising new routes for technological applications in spintronics, especially for memory devices. Systematic experiments using variable photoexcitation wavelength, for example in the THz regime, are desired to

optimize the amplitude of the photoinduced polarization enhancement beyond our proof-of-principle experiment. Finally, given that this effect is confined to the surface region, it should also be effective in the ultrathin limit of a few nanometers, an appealing perspective given the current interest in heterostructure materials.

## Methods

### Time resolved photoemission spectroscopy

The samples were transferred to the tr-ARPES setup using an ultra high vacuum (UHV) suitcase with a base pressure $<1\times10^{-10}$ mbar. All tr-ARPES measurements were performed in UHV at $P<1\times10^{-10}$ mbar and at room temperature, using a laser-based high-harmonic-generation tr-ARPES setup[13] ($h\nu_{probe}=21.7$ eV with p polarization, $h\nu_{pump}=1.55$ eV with s polarization, 500 kHz repetition rate, $\Delta E\sim175$ meV, $\Delta t\sim35$ fs) with a SPECS Phoibos 150 hemispherical analyzer and a 6-axis manipulator (SPECS GmbH). The pump and probe spot sizes (FWHM) are $\sim150\times150\ \mu m^2$ and $\sim70\times60\ \mu m^2$, respectively. All discussed fluences refer to the absorbed fluence, determined using the complex refractive index[30] $n=\sqrt{\epsilon}\sim5.5+4.5i$ at $\lambda=800$ nm.

### Computational details

**Electronic structure calculations.** The presented ab-initio calculations are based on fully relativistic density functional theory as implemented within the multiple scattering Korringa–Kohn–Rostoker Green function based package (SPRKKR)[31]. Relativistic effects such as spin-orbit coupling are treated by Dirac equation. The LDA was chosen to approximate the exchange-correlation part of the potential along with the atomic sphere approximation. The electronic structure is represented using the Bloch Spectral Function (BSF) which consists of the imaginary part of the Green function. In the case of Fig. 1c, a semi-infinite crystal of $\alpha$-GeTe(111) with Te surface termination was considered, as in our previous work[15], representing electronic states of both bulk and surface natures. In order to reproduce the measured photocurrent asymmetry, the one-step model of photoemission[17] was used in order to include all matrix-element effects induced by the experimental geometry: i.e. incoming photon and outgoing photoelectron angles, light polarization and final-state effects. For Fig. 3a–c, an infinite crystal is here treated, accounting therefore only for electronic states of bulk character.

**Drift-diffusion calculations.** The drift-diffusion equations have been solved with a custom code as described in ref. 27 using the parameters listed in the Extended data.

### Samples growth

Ferroelectric $\alpha$-GeTe films (500 nm thick) were grown by molecular beam epitaxy on (111) oriented $BaF_2$ substrates using Ge and Te effusion cells and a growth temperature of 280 °C. Perfect 2D growth was observed by in situ reflection high-energy electron diffraction. After growth, the samples were transferred into a Ferrovac UHV suitcase in which they were transported for the tr-APRES measurements without breaking UHV conditions ($<1\times10^{-10}$ mbar).

### Reporting summary

Further information on research design is available in the Nature Research Reporting Summary linked to this article.

## Data availability

The data that support the findings of this study are available from the corresponding authors upon reasonable request.

## Code availability

The non-equilibrium dynamics of photo-excited carriers data generated in this study from a custom code are available as a .zip file provided as a supplementary data file.

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

## Acknowledgements

G.K, C.W.N. and C.M. acknowledge support from the Swiss National Science Foundation Grant No. P00P2_170597. Ja.M. and L.N. would like to thank the CEDAMNF (Grant No. CZ.02.1.01/0.0/0.0/15_003/0000358) co–funded by the Ministry of Education, Youth and Sports of Czech Republic and the GACR Project No. 20-18725S for funding. This work was supported by the Max Planck Society, the European Research Council (ERC) under the European Union's Horizon 2020 research and innovation program (Grant No. ERC-2015-CoG-682843), and the German Research Foundation (DFG) within the Emmy Noether program (Grant No. RE3977/1). C. Y. and P. W. acknowledge support by SNSF Grant No. 200021-196966. The drift-diffusion calculations have been performed on the Beo05 cluster at the University of Fribourg. G.S. acknowledges support by the Austrian Science Funds, Projects P30960-N27 and I4493-N.

## Author contributions

J.K., G.K., R.E. and C.M. conceived the project. G.S. prepared the samples. Tr-ARPES measurements were performed by G.K., Ju.M., C.W.N., L.R. and C.M. with the help of L.N. and C.S., and were analysed by G.K. Electronic structure calculations were carried out by L.N. and Ja.M. The drift-diffusion calculations have been performed by C.Y. and P.W. The project was coordinated by G.K. together with C.M. J.H.D. and all the authors participated to the interpretation of the data. G.K. wrote the manuscript with input from all authors. L.N. and C.W.N. equally contributed to this work.

## Competing interests

The authors declare no competing interests.
