## [Peer Review File · Nature Communications]

Field-induced ultrafast modulation of Rashba coupling at room temperature in ferroelectric α -GeTe(111)REVIEWER COMMENTS

Reviewer #1 (Remarks to the Author):

The authors present a combined experimental and theoretical study on ferroelectric alpha-GeTe(111). In detail they investigated the transient dynamics in the electronic structure of this Rashba system by use of time-dependent ARPES. New and interesting physics has been discovered on a class of semiconductor materials important for spintronic applications. The findings are very promising as this work helps to complete to a certain degree similar research activities which had been performed by several other experimental and theoretical groups in the past. I fully agree with their conclusions, especially because of the fact that ARPES probes the first few atomic layers only and not the corresponding bulk properties.

In my opinion, the paper is of significant interest for a broad scientific community and certainly deserves publication in a high ranking journal like Nature Communications. The paper is clearly written and the careful analysis, both in experiment and theory, is of excellent quality.

Nevertheless, from the theoretical point of view, only band structure calculations combined with drift-diffusion calculations are shown in order to support the experimental findings. I encourage the authors to present in addition static ARPES calculations especially on the distorted structure.

I guess this point needs to be discussed in the manuscript as spectroscopic calculations would significantly improve the manuscript and in consequence would help to pass the bar for Nature Communications.

In conclusion, I consider this paper an important work which I suggest could be published in a revised version in Nature Communications.

Reviewer #2 (Remarks to the Author):

The manuscript reports the dynamics of the electronic band structure of the bulk Rashba system alpha-GeTe(111) in the femto second(fs) time scale. The modulation of the Rashba spin split and its oscillation in the fs time scale by the laser excitation have been successfully observed. The authors interpreted this phenomenon as the ultrafast lattice distortion of the ferroelectric semiconductor caused by the surface photovoltage effect. The quality of the data is sufficiently good and the data analysis seems to be almost reasonable. The theoretical calculation also supports well the results. The observation of ultrafast dynamics of the electronic structure by time-resolved ARPES is one of the new trends in this field. In addition, the electronic structure dynamics of materials that exhibit the bulk Rashba effect with dielectricity might be of interest to non-specialist researchers too. Thus, I would recommend this manuscript be published in Nature Communications if the authors can explain the following points clearly.

<Question>

The authors decided the peak of the electronic population at 1.5 eV above EF as the position of time zero. The starting time (80 fs) of the electronic state change(=CPb shifts) is estimated from this time zero point, and this delay time is the one of the major reasons why

this phenomenon is related to SPV. Why do the authors consider the peak position should be the time zero point though they consider the onset of the CPb shift as the starting point of the electronic structure change? Namely, why the onset of the electronic population change at 1.5 eV above EF is not the time zero of the observation?

Reviewer #3 (Remarks to the Author):

The manuscript by Kremer et al. reported TrARPES studies on α -GeTe(111) thins. The authors observed a light-induced energy shift of the Rashba band, which can be explained by the light-induced Te atoms' movement supported by first-principles calculations and oscillation with the phonon frequency. Based on the light-induced Te atoms' movement, the key findings are obtained: light-enhanced Rashba coupling and ferroelectric polarization, thus opening the route for controlling ferroelectric polarization in α -GeTe(111). Overall, this work is interesting and the key findings are promising. However, there are a few major questions that need to be addressed before this work can be accepted for publication.

1. The conclusion that the observed band shift is from the light-induced Te atoms' movement is based on the calculated EDC with different Te atom's positions resembling the observed band shift in Fig. 3(b,c). However, the calculated energy shift of CPB is much larger than B1, which contradicts the experimental observation in Fig. 2(d,e). This doubts the explanation of the light-induced Te atoms' movement, further hindering the way to light-enhanced ferroelectric polarization. On the other hand, although the band shift oscillation with a phonon frequency might give some clues for the connection between atomic movement and the observed band shift, the vibration mode of the phonon is still not clear and how the coherent phonon modifies the band shift and ferroelectric polarization needs to be clarified.
2. In addition to the light-enhanced ferroelectric polarization, the light-enhanced Rashba coupling can be directly revealed by extracting the Rashba coupling strength from the band splitting without any theoretical assumption. Such analysis should be added to support the light-enhanced Rashba coupling.

Minor points:

1. Is the refractive index in line 171 on page 11 from measurement or literature? It should be clarified or cited properly.
2. The thickness of the sample should be given.
3. The B1 branch of the Rashba band indicated by the black arrow in Fig. 1(b), 2(a) is hard to see for the readers. The 2D curvature in Fig. S1(a) should be added to Fig. 1 or cited in line 43, which will make it easier to follow.
4. To claim the new potential minimum of the excited state ($x = 0.532$ in Fig. 4(b)), the temporal evolution of band shift needs to be extended to a longer time window.
5. The authors proposed a light-enhanced ferroelectric polarization near the sample surface by introducing a SPV field in Fig. 4(b), however, the influence of the original band bending field on ferroelectric polarization should be considered, as this may also lead to the modulated ferroelectric polarization near the surface as compared to the bulk.
6. In the conclusion, the authors provided perspectives on its application in spintronics and memory devices in an atomically thin film (line 159 on page 11), but it seems that the SPV effect will completely change in the atomically thin film because the SPV is from the interplay between the bulk and the surface.

Report of the Referee 1 – NCOMMS-22-15473-A

The authors present a combined experimental and theoretical study on ferroelectric alpha-GeTe(111). In detail they investigated the transient dynamics in the electronic structure of this Rashba system by use of time-dependent ARPES. New and interesting physics has been discovered on a class of semiconductor materials important for spintronic applications. The findings are very promising as this work helps to complete to a certain degree similar research activities which had been performed by several other experimental and theoretical groups in the past. I fully agree with their conclusions, especially because of the fact that ARPES probes the first few atomic layers only and not the corresponding bulk properties.

In my opinion, the paper is of significant interest for a broad scientific community and certainly deserves publication in a high ranking journal like Nature Communications. The paper is clearly written and the careful analysis, both in experiment and theory, is of excellent quality.

[Our answer:] We thank the referee for their careful reading of our manuscript, their positive appreciation of our work and their comments. Below we give a detailed response to the points raised by the referee.

Nevertheless, from the theoretical point of view, only band structure calculations combined with drift-diffusion calculations are shown in order to support the experimental findings. I encourage the authors to present in addition static ARPES calculations especially on the distorted structure.

I guess this point needs to be discussed in the manuscript as spectroscopic calculations would significantly improve the manuscript and in consequence would help to pass the bar for Nature Communications.

[Our answer:] We thank the referee for their suggestion. We have now added in the revised manuscript a static ARPES calculation for the undistorted structure, replacing the previous Figure 1c (BSF calculation). It reproduces well the experimental left/right asymmetry of the spectral weight as well as the maximum of the photoemission intensity experimentally observed at the crossing of the bulk states at the Gamma point.

As suggested by the referee, we also added a new Figure in the supplementary material showing both BSF and ARPES calculations for the less distorted ($x = 0.52$) and more distorted ($x = 0.534$ and $x = 0.54$) structures (new Figure S5). These calculations clearly show the progressive increase of the Rashba splitting of the bulk states when increasing the x value, confirming our conclusions using the band structure calculations for an infinite crystal presented in Figure 3 in the main part of the manuscript.

We are confident that these new calculations further reinforce our conclusions and improve the quality of our manuscript.

In conclusion, I consider this paper an important work which I suggest could be published in a revised version in Nature Communications.

[Our answer:] We thank again the referee for their detailed and constructive report and for the positive assessment of our work.

Report of the Referee 2 – NCOMMS-22-15473-A

The manuscript reports the dynamics of the electronic band structure of the bulk Rashba system α -GeTe(111) in the femto second(fs) time scale. The modulation of the Rashba spin split and its oscillation in the fs time scale by the laser excitation have been successfully observed. The authors interpreted this phenomenon as the ultrafast lattice distortion of the ferroelectric semiconductor caused by the surface photovoltage effect. The quality of the data is sufficiently good and the data analysis seems to be almost reasonable. The theoretical calculation also supports well the results. The observation of ultrafast dynamics of the electronic structure by time-resolved ARPES is one of the new trends in this field. In addition, the electronic structure dynamics of materials that exhibit the bulk Rashba effect with dielectricity might be of interest to non-specialist researchers too. Thus, I would recommend this manuscript be published in Nature Communications if the authors can explain the following points clearly.

[Our answer:] We thank the referee for their careful reading of our manuscript, their comments, and their positive appreciation of our work. Below we give a detailed response to the questioning of the referee.

<Question>

The authors decided the peak of the electronic population at 1.5 eV above E_F as the position of time zero. The starting time (80 fs) of the electronic state change(=CPb shifts) is estimated from this time zero point, and this delay time is the one of the major reasons why this phenomenon is related to SPV. Why do the authors consider the peak position should be the time zero point though they consider the onset of the CPb shift as the starting point of the electronic structure change? Namely, why the onset of the electronic population change at 1.5 eV above E_F is not the time zero of the observation?

[Our answer:] We thank the referee for their question. Time zero is determined as the time delay when the pump and the probe pulses hit the sample at the same time.

A way to determine time zero is to plot the first observable signal in the unoccupied states which corresponds to the optical excitation of electrons from the top of the valence band (few tens of meV below E_F in the present case) to the conduction band with 1.55 eV photons energy (from the pump). These high energy electrons populate the conduction band, quickly lose energy and redistribute to the bottom of the conduction band (see Figure R1 below). The asymmetry to the right of the peak (see red curve) is due to the finite lifetime of excited electrons in the box of integration.

Figure R1 : (a) Time integrated ARPES spectrum in the first hundreds femtoseconds after time zero showing the optical transition from the valence band (VB) to the conduction (band) induced by the pump pulses. (b) Temporal evolution of CP_B extracted from tr-ARPES measurements taken at an absorbed pump fluence of 1.5 mJ/cm² (blue curve). The red curve corresponds to the transient population of the conduction band and the black curve to the corresponding Gaussian fit.

That is why the photoemission intensity 1.55 eV above the top of the valence band is a good way to determine time zero and also a good estimation of the cross correlation between the pump and the probe pulse (by fitting the rising edge with a Gaussian curve). This is the standard way to do: see Figure 5 in PHYSICAL REVIEW B 105, 075417 (2022) as an example.

We have clarified this point in the revised manuscript by adding the following sentence in the manuscript:

« It corresponds to the first observable signal due to the optical excitation of electrons from the top of the valence band to the conduction band. In these high energy states, the lifetime of the photoexcited electrons is very short and the corresponding trace almost corresponds to the cross-correlation between the pump and the probe pulses (black gaussian curve). The asymmetric shape of the red curve is associated to a non-negligible lifetime with some exponential decay. »

Report of the Referee 3 – NCOMMS-22-15473-A

The manuscript by Kremer et al. reported TrARPES studies on α -GeTe(111) thins. The authors observed a light-induced energy shift of the Rashba band, which can be explained by the light-induced Te atoms' movement supported by first-principles calculations and oscillation with the phonon frequency. Based on the light-induced Te atoms' movement, the key findings are obtained: light-enhanced Rashba coupling and ferroelectric polarization, thus opening the route for controlling ferroelectric polarization in α -GeTe(111). Overall, this work is interesting and the key findings are promising. However, there are a few major questions that need to be addressed before this work can be accepted for publication.

[Our answer:] We thank the referee for their careful reading of our manuscript and their constructive comments about our work. Below we give a detailed response to the questions raised by the referee.

1. The conclusion that the observed band shift is from the light-induced Te atoms' movement is based on the calculated EDC with different Te atom's positions resembling the observed band shift in Fig. 3(b,c). However, the calculated energy shift of CPB is much larger than B1, which contradicts the experimental observation in Fig. 2(d,e). This doubts the explanation of the light-induced Te atoms' movement, further hindering the way to light-enhanced ferroelectric polarization. On the other hand, although the band shift oscillation with a phonon frequency might give some clues for the connection between atomic movement and the observed band shift, the vibration mode of the phonon is still not clear and how the coherent phonon modifies the band shift and ferroelectric polarization needs to be clarified.

[Our answer:] The goal of our DFT calculations of a bulk structure is to present a minimal electronic band structure that reproduces qualitatively well the effect observed in the ARPES measurements with reasonable structural parameters. Still, comparison of such dispersions to experimental data is not trivial and we believe that one cannot expect a perfect quantitative matching with the experimental data, even for the equilibrium case. Therefore, while we agree with the referee that there might be some disagreement between experiment and calculations on the quantitative level, we argue that our calculations qualitatively perfectly support our conclusion that the structural change of the ferroelectric distortion is responsible for the electronic change of the Rashba-like dispersions. We believe therefore that the Rashba splitting of the bulk states is a good indicator of the structural ferroelectric distortion since the ferroelectric distortion itself is at the origin of the Rashba physics in the system (as illustrated in <https://doi.org/10.1038/s41524-020-0274-0> for GeTe and SnTe in paraelectric (undistorted) and ferroelectric (distorted) phases).

As noted by the referee, another strong indication of the link between the enhancement of the Rashba splitting and the light-induced Te atoms' movement is the observation of a coherent phonon oscillation that corresponds to the amplitude mode of the ferroelectric state, i.e. the coherent motion of the Te planes along the FE distortion coordinate.

The coherent excitation of such an amplitude mode is nowadays often observed in time-resolved studies, and in particular in time-resolved ARPES with the corresponding observation of a coherent modulation of the electronic structure. It is then a common approach to model this effect in ARPES by calculating electronic band structures with DFT while imposing atomic displacements that go along the phonon polarization vector, see for instance PRL 108, 256808 (2012), PRL 112, 207001 (2014), PRB 94, 161113(R) (2016) or Communications Physics 2:115 (2019).

2. In addition to the light-enhanced ferroelectric polarization, the light-enhanced Rashba coupling can be directly revealed by extracting the Rashba coupling strength from the band splitting without any theoretical assumption. Such analysis should be added to support the light-enhanced Rashba coupling.

[Our answer:] We thank the referee for their suggestion. We have added in the text an experimental estimation of the Rashba parameter for the ground state case ($t = -200$ fs) and for the time delay of $+200$ fs for which the splitting of the bands is maximised. These values equal $3.1 \text{ eV}\cdot\text{\AA}$ and $3.8 \text{ eV}\cdot\text{\AA}$ respectively, according to the expression $\alpha_R = 2 \cdot E_R / k_0$ (as defined in the Figure R2 below using high resolution ARPES measurement for clarity but also using the tr-ARPES measurements presented in the text) with E_R equal to 170 meV and 210 meV respectively and $k_0 = 0.11 \text{ \AA}^{-1}$ in both cases as shown in the Figure R2 below and the additional Figure S3 included now in the extended data.

It corresponds to a 18% increase of the Rashba parameter, which is of the same order of magnitude as the increase of the lattice distortion we have extracted from the calculations.

We have added the corresponding sentence in the revised manuscript and added a new Figure S3 in the extended data. The corresponding passage in the revised manuscript reads as:

« We have experimentally extracted the transient evolution of the Rashba coupling by estimating the Rashba parameter of the bulk bands at time delays of -200 fs and 200 fs (see Fig. S3). The Rashba parameter is given by the relation $\alpha_R = 2 \cdot E_R / k_0$, where E_R is the energy difference between the CP_B and the top of the B_2 branch, and k_0 their momentum difference. At time delays of -200 fs and $+200$ fs, it equals $3.1 \text{ eV}\cdot\text{\AA}$ and $3.8 \text{ eV}\cdot\text{\AA}$ respectively, corresponding to a 18% increase of the Rashba coupling. ».

Figure R2 : (top) Fermi divided high-resolution static ARPES spectrum of K / GeTe(111) obtained at $h\nu = 21.2$ eV and $T = 300$ K. (bottom) tr-ARPES spectra recorded at pump-probe delays of -200 fs (left) and $+200$ fs (right) showing the transient enhancement of the Rashba parameter.

Minor points:

1. Is the refractive index in line 171 on page 11 from measurement or literature? It should be clarified or cited properly.

[Our answer:] The refractive index has been determined by using experimental parameters obtained from the literature. We thank the referee for noticing this point and we have added a proper citation (*Nature Mater* **7**, 653–658 (2008)).

2. The thickness of the sample should be given.

[Our answer:] The thickness of the MBE grown samples corresponds to 500 nm and has been added to the methods part of the revised manuscript.

3. The B1 branch of the Rashba band indicated by the black arrow in Fig. 1(b), 2(a) is hard to see for the readers. The 2D curvature in Fig. S1(a) should be added to Fig. 1 or cited in line 43, which will make it easier to follow.

[Our answer:] We understand the point raised by the referee and we have added a reference to Fig. S1(a) in line 43 of the main text.

4. To claim the new potential minimum of the excited state ($x = 0.532$ in Fig. 4(b)), the temporal evolution of band shift needs to be extended to a longer time window.

[Our answer:] Unfortunately, we do not have high quality ARPES data in a longer time window which could be used to extract the evolution of the CP_B to longer time delays. After half of the period starting from the maximum amplitude of the oscillation i.e. at $t = 200 + 100 = 300$ fs, we obtained a shift of CP_B corresponding to 20 meV which corresponds well (using a linear approximation) to an intermediate value between $x = 0.534$ (40 meV shift) and $x = 0.53$ (0 meV shift: the reference point). This is the reason why we claim that in a harmonic potential picture, the new minimum is localized at $x = 0.532$.

The most important effect on the electronic band structure happens in the first hundreds femtosecond (especially the delayed shift of the CP_B) and is sufficient, from our point of view, to capture the physics and explains the underlying mechanism of the field induced modulation of Rashba coupling in GeTe(111).

5. The authors proposed a light-enhanced ferroelectric polarization near the sample surface by introducing a SPV field in Fig. 4(b), however, the influence of the original band bending field on ferroelectric polarization should be considered, as this may also lead to the modulated ferroelectric polarization near the surface as compared to the bulk.

[Our answer:] The effect on the ferroelectric polarization of the original band bending field is implicitly included in the observed Rashba splitting by ARPES in the ground state. This is the initial state of our system and it cannot be separated from the field induced by the lattice distortion of the crystal.

What we can say in our case is that the photoinduced compensation of this original band bending field leads to a coherent movement of the lattice which further increases the lattice distortion, the total electric field and the Rashba splitting in the system.

The effect of the initial band bending electric field could be, in principle, observed in the paraelectric phase. This is a high temperature phase in a case of GeTe which is not accessible with ARPES measurements ($T = 700$ K).

6. In the conclusion, the authors provided perspectives on its application in spintronics and memory devices in an atomically thin film (line 159 on page 11), but it seems that the SPV effect will completely change in the atomically thin film because the SPV is from the interplay between the bulk and the surface.

[Our answer:] In the conclusion, we have mentioned the ultrathin limit of a few nanometers but not the atomically thin limit. We are thinking in particular of thicknesses of about ten nanometers, which have been used recently in devices for spintronics measurements and for which we can still expect to have SPV effects (reference 9 in the manuscript)

REVIEWERS' COMMENTS

Reviewer #2 (Remarks to the Author):

The authors answered my questions in detail and properly and added some new sentences to clarify the point in the revised manuscript.

Thus, I think the revised manuscript is ready to be accepted in this form.

Although this is not the referee's job I think, I found several typos in the manuscript as follows. I recommend the author do spellcheck once.

line 40, with with >>> with

line 62 >>>approximately

line 90 >>>paraelectric

line119 >>>diminution

line122 >>>displacive

Reviewer #3 (Remarks to the Author):

The authors have addressed my questions and improved the manuscript. I would recommend publication.

Report of the Referee 2 – NCOMMS-22-15473-A

The authors answered my questions in detail and properly and added some new sentences to clarify the point in the revised manuscript. Thus, I think the revised manuscript is ready to be accepted in this form. Although this is not the referee's job I think, I found several typos in the manuscript as follows. I recommend the author do spellcheck once.

line 40, with with >>> with
line 62 >>>approximately
line 90 >>>paraelectric
line119 >>>diminution
line122 >>>displacive

[Our answer:] We thank the referee for their careful reading of our manuscript and for their positive appreciation of our work. We have corrected the different typos and spellcheck again our manuscript.

Report of the Referee 3 – NCOMMS-22-15473-A

The authors have addressed my questions and improved the manuscript. I would recommend publication.

[Our answer:] We thank again the referee for their careful reading of our manuscript, and for their positive recommendation for publication.